# Genomic Prediction in Local Breeds: The Rendena Cattle as a Case Study

**DOI:** 10.3390/ani11061815

**Published:** 2021-06-18

**Authors:** Enrico Mancin, Beniamino Tuliozi, Cristina Sartori, Nadia Guzzo, Roberto Mantovani

**Affiliations:** 1Department of Agronomy, Food, Natural Resources, Animals and Environment, University of Padua, Viale dell’Università, 16, 35020 Legnaro, Italy; beniamino.tuliozi@unipd.it (B.T.); cristina.sartori@unipd.it (C.S.); roberto.mantovani@unipd.it (R.M.); 2Department of Comparative Biomedicine and Food Science, University of Padua, Viale dell’Università, 16, 35020 Legnaro, Italy; nadia.guzzo@unipd.it

**Keywords:** local cattle breeds, genomic prediction, ssGBLUP, cross-validation

## Abstract

**Simple Summary:**

Although genomic selection is being used in many livestock species, it has not yet been considered in local breeds due to the lower population size and the potential less effective impact on the genetic evaluation of these breeds. The current research aims to investigate how genomic data can impact the accuracy of genetic predictions for beef traits in Rendena, a small local cattle breed of the North-East of Italy selected for a dual purpose. Classical animal models using only phenotypic information were compared with two models that integrated genomic data with pedigree information. The genomic models presented better accuracy in estimated breeding values of the animals than the ‘classical’ animal model, especially the ‘simpler’ one assuming homogeneous variances of single nucleotide polymorphisms. Our results show that the inclusion of genomic information can be successfully applied to breeding selection scenarios even in small local cattle breeds such as Rendena.

**Abstract:**

The maintenance of local cattle breeds is key to selecting for efficient food production, landscape protection, and conservation of biodiversity and local cultural heritage. Rendena is an indigenous cattle breed from the alpine North-East of Italy, selected for dual purpose, but with lesser emphasis given to beef traits. In this situation, increasing accuracy for beef traits could prevent detrimental effects due to the antagonism with milk production. Our study assessed the impact of genomic information on estimated breeding values (EBVs) in Rendena performance-tested bulls. Traits considered were average daily gain, in vivo EUROP score, and in vivo estimate of dressing percentage. The final dataset contained 1691 individuals with phenotypes and 8372 animals in pedigree, 1743 of which were genotyped. Using the cross-validation method, three models were compared: (i) Pedigree-BLUP (PBLUP); (ii) single-step GBLUP (ssGBLUP), and (iii) weighted single-step GBLUP (WssGBLUP). Models including genomic information presented higher accuracy, especially WssGBLUP. However, the model with the best overall properties was the ssGBLUP, showing higher accuracy than PBLUP and optimal values of bias and dispersion parameters. Our study demonstrated that integrating phenotypes for beef traits with genomic data can be helpful to estimate EBVs, even in a small local breed.

## 1. Introduction

Rendena is a dual-purpose cattle breed indigenous to the North-East of Italy. This breed is included within the “European Federation of Cattle Breeds of the Alpine System” (FERBA), an organization whose main purpose consists in the preservation and promotion of local cattle breeds of the alpine system (http://www.ferba.info, accessed on 20 April 2021). As is the case with many indigenous breeds, a greater genetic diversity than specialized and cosmopolitan breeds is expected also for the Rendena [1]. This remarkable biodiversity is of great ecological importance and can be a beneficial factor for the survival of the local population. Moreover, traditional breeds such as Rendena provide additional benefits to the local human population such as economic advantages, ecosystem services, and also cultural benefits, such as preservation of cultural heritage and tradition of a specific area [1]. Rendena cattle also shows excellent values for traits concerning fertility and longevity, maintains a median milk production (5000 kg per lactation), and possesses a fairly good beef conformation [2]. Rendena cows are selected for both milk and meat, but with more emphasis on dairy production in the selection index [3], with dairy accounting for 65% and beef traits for 35% [4]. Although beef attitude plays a less important role than milk in the selection index, an increase in the accuracy of the selection for this feature over time could prevent its detriment due to the antagonism with milk production [3]. Estimations of breeding values (EBVs) have until now mostly taken place using classical animal model analysis in Rendena through best linear unbiased predictor (BLUP; [3]) for traits related to milk, meat production, and linear type traits. However, several studies have shown how the use of genomic data can lead to an increase in prediction accuracy compared to using only pedigree information [5].

For a long time, two major limitations to the genomic selection approach on small populations such as Rendena have been the prohibitive cost of genotyping a sufficient number of single nucleotide polymorphisms (SNPs) per individual and the equations for EBVs’ estimation, which were based on a multistep approach [6]. In fact, the drawback of the multistep approach in small populations is the scarce number of genotyped animals with a phenotype to be used as the reference population to ensure a good accuracy of prediction [6]. This is even more noticeable when sex-limited traits are considered [7]. To overcome this problem, methods such as the use of de-regressed proof [8,9] have been developed to allow the inclusion of animals whose only genotype is known, using progeny yield deviation adjusted for mates as a pseudo-phenotype. However, this method presented some biases and lower accuracy whenever animals have few progenies with the phenotype [10,11].

However, in the last few years, both limitations preventing the use of the genomic selection approach in small breeds with limited diffusion such as Rendena have subsided. Firstly, the constant decline in prices of SNP platforms has allowed genomic selection to become much more cost-efficient. Secondly, equations such as the single-step genomic best linear unbiased prediction (ssGBLUP) have been developed and found to be suitable even in small-breed contexts [12]. The ssGBLUP simultaneously evaluates genotyped and non-genotyped animals by substituting the pedigree-based relationship matrix (A) present on BLUP, with a relationship matrix that combines pedigree and genomic information, usually called H [13]. Single-step GBLUP represents a simple alternative to de-regressed proofs. Moreover, ssGBLUP offers the advantage of avoiding double counting contributions, and it implicitly limits the bias of preselection for genotyped animals without the phenotype [14,15,16]. Several studies have shown that ssGBLUP outperformed other methods in different livestock species in the context of genomic selection [17]. On the other hand, ssGBLUP might have its own drawback: the genomic relationship matrix (G) included in a single step assumes that all SNPs explain the same amount of variance [18]. This may be a limit in the presence of traits influenced by many quantitative traits loci (QTL), such as some beef-related traits such as carcass weight and daily gain [19,20]. Indeed, some studies reported that SNP regression equations, in which prior assumption of SNPs’ effect and variance are modeled with different a priori assumptions, outperformed the prediction of ssGBLUP [21]. On this point, Zhang et al. [22] proposed to “relax” the assumption of the G matrix in which all SNPs equally contribute to the genomic variance of the traits by adding specific SNPs weights. These methods are called weighted single-step GBLUP (WssGBLUP), and in a recent study, it has been shown to be effective by increasing the accuracy with respect to ssGBLUP for phenotypes such as those related to the beef attitude [23]. In this study, we investigated if the inclusion of genomic data in the estimate of breeding values for three key beef traits measured during performance tests in Rendena might increase their predictive accuracy with respect to traditional pedigree BLUP (PBLUP). In particular, the objective of this study was both to test different single-step GBLUP methods for beef traits and to measure their difference in accuracy using alternative weighting strategies, i.e., among different weighted single-step GBLUPs.

## 2. Materials and Methods

### 2.1. Data Availability

#### 2.1.1. Phenotypic and Pedigree Data

Phenotypic and pedigree information was provided by the National Breeder Association of Rendena Cattle (www.anare.it, accessed on 18 March 2021). The phenotypes consisted of data recorded on Rendena young bulls during performance tests conducted from 1985 to the present. The phenotypes used were the average daily gain (ADG) obtained by linear regression of weight on age recorded at least 11 times during the stay of bulls at the test performance test station, the mean in vivo fleshiness score (EUROP grade), and the mean in vivo estimate of dressing percentage (DP) evaluated by three skilled classifiers at the end of the test, i.e., about 11 months of age, distribution of phenotype are reported in Table 1. In vivo fleshiness score (EUROP grade) was linearly transformed as previously reported [4]. In the final dataset, 1691 animals and as many phenotypes were present. The animals present in this dataset were born between 1985 and 2020. In addition, 8372 animals in the pedigree were retrieved tracing back up to the 10th generation.

#### 2.1.2. Genotype Data

Two genotyping platforms were used in this study, Illumina Bovine LD GGP v3, including 26,497 SNP markers (LD; no. = 1427) and Bovine 150K Array GGPv3 Bead Chip, comprising 138,974 SNPs (HD; no. = 554; Illumina Inc., San Diego, CA, USA). The higher density panel was used only in 554 males, whereas the remaining males (no; = 174) and all the females (no. = 1253) were genotyped with the LD platform. Males genotyped with LD chips were all animals with at least one father and one full sib genotyped with the HD chip. The two panels shared about 60% of markers. Females with a call rate (CR) lower than 95% and males with a CR lower than 90% were discarded before the analysis. In addition, for both platforms, SNPs with a minor allele frequency (MAF) <0.01 and call rate lower than 0.90 were removed with the plink program [24]. Before genomic imputation, possible progeny conflicts were corrected with the seekparentsf90 program [25]. The imputation of LD samples to HD density was performed with the AlphaImpute2 program [26], which combines algorithms of population imputation with the use of imputation from pedigree information utilizing a sort of multi-locus iterative peeling [26]. To avoid excessive computational demand, we imputed one chromosome at each time. The threshold of loci inclusion to HD panels was set to 0.90, and a conservative genotype threshold for imputation of 0.99 was chosen. A further genomic quality control was then made for whole imputed panels (1953 individuals): SNPs with MAF lower than 0.05 with Hardy–Weinberg equilibrium lower than 0.15 and a call-rate under 0.90 were removed from the dataset. In addition, animals with a call-rate under 0.90 were removed, in this case, using the preGSf90 program [25]. At the end of genotype editing, 1743 animals were retained for further analysis, 690 of which had both phenotype and genotype, consistency of this information are reported on Figure 1. The genotyped females are close relatives with the male in the performances test, i.e., dams or grad-dams.

### 2.2. Prediction Model

#### 2.2.1. Pedigree Best Linear Unbiased Prediction (PBLUP)

The same fixed and random effects were used for all analyzed traits with the following model:(1)y=Xb+Za+e
where ***y*** is the vector of phenotypes, ***X*** represents the incident matrix for systematic fixed effects, and ***b*** is the vector of fixed effects. Two cross-classified effects were used as in [4]: the contemporary group (142 levels) and the parity order of the cow (four classes: first parity, second parity, third to seventh parity, and above the eighth parity included). ***Z*** is the incident matrix of random genetic additive effects, while a represents the vector of the additive genetic effects (EBVs) and e is the vector of residuals sampled from a distribution $N\left(0,I\sigma_{e}^{2}\right)$**,** where $\sigma_{e}^{2}$ is the residual variance. The additive genetic effect was sampled from a normal distribution with mean zero and variance $\sigma_{a}^{2}$ and a covariance structure depending on the model used. In the PBLUP, the model covariance of the random genetic effect was sampled from a distribution$\;N\left(0,A\sigma_{a}^{2}\right)$, with **A**, which represents the identical by descendent (IBD) matrix constructed from pedigree information. All genetic and genomic prediction models were carried out with the *blupf90* suite of programs [27].

The variance components used in all prediction scenarios were estimated under this model using the univariate approach. In addition, genetic and residual correlation among traits was estimated with multi-traits models. Covariances’ structures were **G**⊗**A** and **R**⊗**I** with **G** and **R,** are 3 × 3 matrices, respectively, including the additive genetic and the residual (co)variances matrices, ⊗ is the Kronecker product, and **A** and **I** are the additive relationships matrix and an identity matrix, respectively. Prior distributions for **G** and **R** matrices were independent inverse Wishart. Genetic and residual correlations (*r_a_*) were calculated between trait pairs as the ratio of the covariance on the square root of the product of the respective variances.

Variances were estimated using Gibbs’s sampling algorithm with gibbs3f90 program [27]. A chain of 200,000 iterations was used in both models. The first 5000 samples were discarded as burn-in. Samples were stored every 100 iterations to leave 1950 samples for inference.

#### 2.2.2. Single-Step Genomic Best Linear Unbiased Prediction (ssGBLUP)

In ssGBLUP, the inverse of the IBS matrix $A^{-1}$ was replaced by the $H^{-1}$ matrix as follows:(2)$$H^{-1}=A^{-1}+\left[\begin{array}{cc} 0 & 0\\ 0 & {\left(\alpha G+\beta A_{22}\right)}^{-1}-A_{22}^{-1} \end{array}\right]$$
where $A^{-1}$ and $A_{22}^{-1}$ represent the inverses of the IBD matrix for all individuals and only genotyped animals, respectively. To avoid singularity problems, the bending coefficient *α* and *β* were set to 0.95 and 0.05, respectively. $A^{-1}$ was computed accounting for inbreeding to avoid inflation (bias) and to reduce the distance between the two matrices, as suggested elsewhere [28]. G is the genomic relationship matrix, built using the first method proposed in [18]:(3)$$G=\frac{MM^{\prime }}{2\sum p_{i}\left(1-p_{i}\right)}$$
where *M* is a matrix of SNP content centered by twice the current allele frequencies, and pi is the allele frequency for the ith SNP. In addition, variance components were re-estimated under this model to evaluate variances changes by the inclusion of genomic data. Therefore, *G* in ssGWAS is adjusted so that the average diagonal and off-diagonal matches the averages of A22.

#### 2.2.3. Weighted Single-Step Genomic Best Linear Unbiased Prediction (WssGBLUP)

The last method used for genetic prediction was WssGBLUP, which differs from ssGBLUP in the construction of G. Particularly, the ***G_w_*** matrix was built using the following method [17]:(4)$$G_{w}=\frac{MDM^{\prime }}{2\sum p_{i}\left(1-p_{i}\right)}$$
where D is a diagonal matrix in which the elements of the diagonal correspond to the weight or effect of each SNP. Generally, SNPs’ effects $\left(\hat{u}\right)$ are obtained as a function of the SNPs effect through a back-solving procedure from the EBVs’ solution obtained iteratively with the (W)ssGBLUP [29] as follows:(5)$$\hat{u}=\delta \alpha \frac{1}{2\sum p\left(1-p\right)}DM^{\prime }{\left[MDM^{\prime }\right]}^{-1}\hat{a}$$
where $\hat{a}$ is the vector of solutions of the genomic breeding values of the genotyped animals, and *δ* accounts for the difference in genetic base between the pedigree and genomic relationship.

An iterative algorithm following that reported in [16] was used. This algorithm consists of the subsequent steps:

Initial parameters are set to $t=1,D_{\left(t\right)}=I,G_{\left(t\right)}=\frac{MD_{\left(t\right)}M^{\prime }}{2\sum p_{i}\left(1-p_{i}\right)}$.GEBV ($\hat{a}$) is obtained using ssGBLUP algorithm.Allele substitution effects for each SNP ($\hat{u}$) are reported in [5] with *postGSf90* [22].Each $d_{i\left(t+1\right)}$ element of $D_{\left(t+1\right)}$, such as $CT^{\frac{\left|{\hat{u}}_{i}\right|}{sd\left(\hat{u}\right)}-2}$, is then calculated as in [18], where *CT* is a shrinkage factor determining how much the distribution of SNP effects departs from normality.SNP weights are normalized by keeping genetic variance constant among iteration:$$D_{\left(t+1\right)}=\frac{tr\left(D_{\left(1\right)}\right)}{tr\left(D_{\left(t+1\right)}\right)}tr\left(D_{\left(t+1\right)}\right).$$*G* is then re-built with the new obtained weights as $G_{\left(t+1\right)}=\frac{MD_{\left(t+1\right)}M^{\prime }}{2\sum p_{i}\left(1-p_{i}\right)}$.Further iterations are carried out up to convergence using WssGBLUP.

#### 2.2.4. Weighted Strategies

A further aim of this study was to identify optimal weight strategies to achieve higher accuracy and less biased genomic prediction. *NonlinearA* methods [18,30] were used as the weighting strategies. We focus on the effect of variance limitations (*limit*) and the shrinkage factor (*CT*). Other strategies such as linear weight [22,31] or Bayesian variable selection methods [32] were not applied in this study because of their excessive shrinkage that led to high biased prediction and incompatibility between the **A** and **G** matrix, as reported in Appendix A.

Three values of *CT* were used in this study (1.105, 1.125, and 1.250), considering that values greater than one deviate proportionally from a normal distribution and exhibit grater shrinkage. By default, the *postGSf90* program set maximum change in SNPs variance equal to $CT^{\left(5-2\right)}$; thus, default limitations for the three parameters were automatically set to 1.350, 1.424, and 1.953 for *CT* equal to 1.105, 1.125, and 1.250. Other scenarios have been explored, setting the maximum change on variance equal to 5.

### 2.3. LR Cross-Validations

Estimators of bias, dispersion, and accuracy were adopted to evaluate the different prediction models. The LR cross-validation method was used on this behalf [33]. In this approach, two datasets (whole and partial) were used, and the parameters described above were estimated in a set of focal individuals. The whole dataset contains all populations’ information, while the partial dataset includes a subset of phenotypic data up to a given date. In this study, 2015 was set as the cut-off year, and the focal individuals are the younger bulls with only genotype information (i.e., born after 2015; 109 animals). The focal individuals represented the young animals of interest for selection, and in most of the cases, they represented the young “genomic” candidate for selection [33]. Simply speaking, focal individuals are the animals for which accuracy of prediction is of greater interest for selection.

LR defined bias as $bias=\bar{{\hat{u}}_{p}}-\bar{{\hat{u}}_{w}}$, where ${\hat{u}}_{p}$ is the estimate of individual EBV in the partial dataset and ${\hat{u}}_{w}$ is the estimate of individual EBV in the whole dataset. Bias equal to 0 stands for unbiased prediction. Due to the different magnitudes of each trait, bias was also standardized by the genetic standard deviation of each trait analyzed.

Dispersion was described by the slope of the regression between EBVs in the whole dataset to EBVs in the partial one, i.e., $disp=\frac{cov\left({\hat{u}}_{w,}{\hat{u}}_{p}\right)}{var\left({\hat{u}}_{p}\right)}$, with an expectation of 1, i.e., *disp* <1 designate over-dispersion, while *disp*>1 indicates an under-dispersion.

In this study, we describe as accuracy (*acc*) the correlation of breeding values estimated in the two datasets [33]: $acc=\frac{cov\left({\hat{u}}_{w,}{\hat{u}}_{p}\right)}{\sqrt{var\left({\hat{u}}_{p}\right)var\left({\hat{u}}_{w}\right)}}$. This estimator stands for the inverse of accuracy gain when the phenotype was added, moving from the partial dataset to the whole one. Low values of the “*acc*” estimator mean that the EBV estimate of the focal group is mainly influenced by the addition of new phenotypic information with respect to the conditional kinship information. Thus, $E\left(acc\right)\approx \frac{acc_{p}}{acc_{w}}$.

Furthermore, reliability, the squared accuracy, was obtained through the following approximated expression: $rel=\frac{cov\left({\hat{u}}_{w},{\hat{u}}_{p}\right)}{\left(1-\hat{F}\right)\sigma_{u}^{2}}$, where $\hat{F}$ is the average population inbreeding coefficient and $\sigma_{u}^{2}$ is the genetic variance estimated in the whole dataset. The expected value for rel is equal to $acc^{2}$, and the adequacy of this estimator was proofed in Appendix 1 in [34]. Note that no differences were observed in terms of variance components between the whole dataset and the focal groups; thus, for that reason, adjustment by selected reliability proposed in [35] was not applied.

In addition, according to [34], the increase in accuracy when genomic data are introduced was estimated as $inc=\rho_{A,G}{}^{-1}-1$, where $\rho_{A,G}=\frac{cov\left({\hat{u}}_{A,}{\hat{u}}_{G}\right)}{\sqrt{var\left({\hat{u}}_{A}\right)var\left({\hat{u}}_{G}\right)}}$, ${\hat{u}}_{A}$ is the EBV estimated with PBLUP in the partial dataset and ${\hat{u}}_{G}$ is EBV estimated using genomic information in the partial dataset. In fact, using the same reasoning done for *acc,* $\rho_{A,G}$ quantifies the increase of the inverse of accuracy when genomic data are added, because its expected value is $\frac{acc_{A}}{acc_{G}}$. A further evaluation of the increase in accuracy due to genomic data was also obtained following [34], which suggested adjusting the increase in accuracy for the ratio of genetic variances of two models accounting or not for genomic information, i.e., $inc_{adj}=\frac{\sigma_{A}^{2}}{\sigma_{G}^{2}}inc$, where $\sigma_{A}^{2}$ is the genetic variance estimated with only pedigree information and $\sigma_{G}^{2}$ is the variance when genomic information is included. For the matter of simplicity, only *inc_adj* has been reported as a parameter that identifies the increase of accuracy. Note that EBV in the focal populations is normally distributed; thus, conditions under the LR assumption were not violated.

## 3. Results

### 3.1. Variance Components

Heritability (h^2^) and genetic and residual correlations estimated using PBLUP are reported in Table 2. All traits presented a medium to high heritability. EUROP was the trait with lowest heritability, 0.304, while ADG and DP showed an h^2^ of 0.335 and 0.392, respectively. In addition, all traits’ pairs, as expected, presented medium to high genetic and residual correlations. ADG presented a medium-positive genetic correlation with the other two traits (0.38 on average), while DP and EUROP were strongly correlated (0.981) to be considered a unique trait.

Table 3 reported estimated heritability and genetic and residual correlations using ssGBLUP. In this case, both h^2^ and correlations had similar results to those estimated with the PBLUP. For what concerns h^2^, ADG decreased by about 0.02, while EUROP increased by about 0.04 in ssGBLUP as compared to PBLUP. On the other hand, DP remained basically unchanged comparing the two approaches. Correlations presented almost the same values in both analyses, with the only exceptions of the genetic and residual correlations between ADG and EUROP that resulted in an increase in ssGBLUP of about 0.02 and 0.08, respectively.

### 3.2. Weighting Strategies

Figure 2 shows how different values of *CT* and the limitation of SNPs’ variance can affect genomic prediction.

As expected, higher accuracy (Figure 2A) was reached in the WssGBLUP analyses with the increase of the number of iterations, although in most cases, the asymptote was reached at the second iteration, with the only exception of the CT 1.25 with the limit of maximum variance established at 5, which reached the maximum accuracy after 3–4 iterations. Variance limits did not affect accuracy using a CT of 1.105 or 1.125.

Bias (Figure 2C) followed the same trends in all phenotypes; at first, iteration bias was even lower than with ssGBLUP, but when iterations increased, bias rapidly increased. ADG presented higher biases, even if the difference in magnitude was considered by standardizing values obtained. Even dispersion (spread; Figure 2B) followed the same trends as accuracy with an increase after 2/4 iterations depending mainly on the value attributed to CT. For all traits, as the interactions increased, dispersion departed from the expected value of 1, although for EUROP, the use of CT at 1.125 was maintained steadily close to 1.

In general, higher CT values (that is, greater departures from normality) presented better accuracy but more under-dispersion and biases. When CT changed from 1.105 to 1.125, accuracy increased by 2% in all phenotypes, and a substantial increase in accuracy was observed, moving to a CT value of 1.250 (+20% on average). When the threshold for maximum SNPs variance was raised up to 5, accuracy increased slightly, especially from the third to tenth iteration.

Figure 3, Figure 4 and Figure 5 show the percentage of variance explained by a sliding window of 20 non-overlapping SNPs. These plots show how the different values of CT and limit influenced the shrinkage SNPs. Furthermore, observing the peaks in the Manhattan plots, it can be seen how these traits are potentially controlled by few QTLs. The high peak found on chromosome 22 for EUROP and DP can explain why these traits are highly genetically correlated.

### 3.3. Model Comparison

From the previous analysis, for each phenotype, two weighting strategies were retrieved: the one presenting a value of bias close to the optimal value (WssGBLUP_1) and the one with highest accuracy (WssGBLUP_2) The weighting strategies that produced the lowest bias were associated with a CT = 1.105, default value for limit, and iteration 1. On the other hand, as reported previously, CT = 1.250 and limit equal to 5 produced the best results in terms accuracy of prediction. For ADG and DP, maximum accuracy value was found on iteration 4, while for EUROP, iteration 7 was the most successful (Figure 2). Table 4 shows the different performances of prediction of PBLUP, ssGBLUP, and the two selected WssGBLUP obtained under the LR cross-validation method. EUROP presented the highest accuracy in all models considered, followed by DP and ADG. All traits presented a bias value close to 0, although DP presented a slightly positive bias of about 0.02 on average. Generally, all models except WssGBLUP_2 showed very low biased prediction, also considering that estimated genetic progress per year is consistent, being positive and equivalent to 0.58, 0.42, and 0.33 standard deviations for ADG, EUROP, and DP, respectively (Figure 6).

For what concerned the dispersion parameter, in this study, we found that ADG and DP were slightly under-dispersed, while EUROP was a little over-dispersed for PBLUP and WssGBLUP_1, showing values of dispersion <1.

When only pedigree information was used, lower accuracy was observed for all traits: ADG presented a value of 0.366, EUROP of 0.464, and DP of 0.506. Lower reliability values were also found in this model. Interesting, PBLUP is the only prediction model in which a marginally negative bias was observed. PBLUP presented similarly biased values as WssGBLUP_1 and ssGBLUP for EUROP and DP (same absolute value but opposite sign), while for ADG, it presented a greater absolute value of bias than for the other two methods.

When genomic information was added, a global increase in accuracy and reliability was observed. In ssGBLUP models, increases in accuracy of 0.106, 0.087, and 0.064 were observed for ADG, EUROP, and DP, respectively. Reliability estimators showed the same trend. The ssGBLUP had higher accuracy values with respect to PBLUP, and it also presented bias and dispersion closest to optimal value. As can be seen from Figure 2 and Table 4, higher accuracy and reliability values were observed as SNPs shrinkage increased (that is, for higher values of CT); however, in parallel, more under-dispersion and biased predictions were found.

The inc_adj estimator represents the increase of accuracy when genomic models were used. ADG is the trait that was most favored from the introduction of genomic data, with a value of 45%, and DP is the one with lower benefits (26.7%). WssGBLUP_1 presents a similar inc_adj value than ssGBLUP, while in WssGBLUP_2, value rises to 4 percentage points in ADG as well as to 5 and 7 percentage points in EUROP and DP.

## 4. Discussion

In this study, we evaluated how the use of genomic data can improve the estimates of breeding values in the local dual-purpose Rendena cattle. We used data relative to beef traits collected in the performance tests of young bulls, both because these traits are accounted for in the selection index and also because of the smaller number of genotyped individuals needed with respect to traits such as milk production. In addition, this was the first approach to apply genomic selection in a small local cattle breed.

The three performance test phenotypes i.e., ADG, EUROP, and DP, presented medium to high heritability, ranging from 0.30 to 0.40. We recorded little difference from the study of [4], even if our dataset was greater by an amount of about 40%. The heritabilities of these traits were similar to the ones observed in other dual-purpose (i.e., Alpine Grey; [36] or beef specialized breeds [37,38,39]). All traits appeared highly genetically correlated, especially EUROP and DP, as expected and widely reported in the literature [38,40,41]. Interestingly, after we introduced genomic data, we did not observe many discrepancies in terms of genetic (co)variance(s) with respect to those estimated with PBLUP. This is in agreement with what was reported in [42], i.e., that even for non-random genotyping strategies, the population variances in ssGBLUP are not influenced by the selective genotyping strategies as much as they are with GBLUP [43]. In fact, thanks to the contribution of non-genotyped animals present in the pedigree, the bias due to the preselection of genotyped animals in ssGBLUP is reduced. Furthermore, the genotypes were homogeneously distributed over years, and this factor may have undoubtedly contributed to reducing discrepancy in terms of variance estimates.

The usefulness of genomic selection was assessed using LR as a cross-validation method, which provided accuracies, bias, and dispersion of the genetic evaluations. LR presents several advantages [33]: the robustness of genetic evaluations is inferred on a target group of animals, i.e., accuracy can be evaluated at the level of the preferred sub-group of the population. In our study, our focus was on young bulls and close relatives, the sub-group in which phenotypic data were collected. In addition, another advantage consists of the fact that LR does not require the precorrection of phenotypes, thus avoiding potentially biased prediction due to the heterogeneity of the contemporary groups (number of animals range from 4 to 20 animals per group [33]).

Results confirmed that when genomic data were integrated with pedigree, there was a substantial increase in the accuracy of (G)EBVs prediction. Accuracy increased by about 30% on average when switching from BLUP to ssGBLUP. Moreover, an additional increase in accuracy was observed when weighting strategies were applied, i.e., from 0.366 to 0.472 for ADG, from 0.509 to 0.569 for EUROP, and from 0.464 to 0.528 for DP, respectively. These outcomes suggest that the genomic information can potentially capture variation in Mendelian sampling, thus leading to a greater accuracy of prediction when only kinship information is used [44]. A similar impact of ssGBLUP on the accuracy of performance test traits has been observed in Hanwoo beef cattle [45], in which the same number of phenotypes and genotypes were used; however, results cannot be compared numerically due to the different cross-validation strategies implemented.

The findings of previous studies report that the ssGBLUP led to more accurate predictions than the BLUP. Other research conducted on a different type of beef-related traits presented a substantial increase in breeding value prediction when ssGBLUP was used. However, those investigations were conducted with breeds with much larger population sizes, and results were expressed in terms of reliability [46,47,48]. Interestingly, in Cesarani et al. [49], an analogous number of animals was used, and results in terms of bias and dispersion agree with results obtained in this manuscript, with a similar influence of weighting strategies, although the number of animals with genotype in their study is much lower than ours. While generally, different weighting strategies have led to different increases in the accuracy of the breeding value predictions [21], extreme shrinkage strategies (i.e., quadratic weight) can lead rapidly to a decline in accuracy as the interactions increase and generally present greater biased prediction [23]. These weighting strategies have thus been discarded from this study due to the excessive shrinkage caused by the influence of major QTLs (Appendix A). In addition, an extreme shrinkage can lead to an incompatibility between G and A matrix, consequently losing some properties of the single step such as unbiasedness of selection.

For that reason, nonlinearA methods were chosen over the other weighting strategies. The consistency of nonlinearA methods in a single-step framework has been reported by [30]. The augment of the accuracy of weighting strategies is particularly relevant when small datasets are used, such as in the present study [50]; moreover, the use of heterogeneous SNP weighting is useful when the number of SNPs exceeds the number of animals [50]. This point could be relevant for our study, since redundant information can be produced also by the genomic imputation [51]. In fact, according to [52], when the trait is controlled by a few QLTs and few genotyped animals are present, the information relative to the trait in the genome is usually divided into few blocks, and consequently most of the SNPs information is considered redundant. Assigning different values to SNPs or to chromosome segments can remove redundant SNPs information [53]. The presence of major QTLs has a positive impact on WssGBLUP, because the relationships between animals are focused on SNPs, which are clearly linked to the QTLs [54].

Despite this, LR cross-validation methods pointed out that major under-dispersion and bias are observed by applying WssGBLUP. In our study population, proven and/or young animals are evaluated with the rest of the performance test animals. The higher bias present in some of the weighting models led to an inaccurate estimation of genetic trends than in turn led a potentially biased selection decision, i.e., selecting only young animals with respect to the older ones [35]. Because of that, the bias and dispersion parameters must be considered alongside the accuracy of selection [55]. For this reason, models over the second iteration can be discarded from the choice of model with “best” properties, due to the lack of mean’s exact estimation in selected animals [33]. Interestingly, a decline in biases was observed in the first iteration for all phenotypes. Conversely, PBLUP and ssGBLUP confirmed their unbiasedness prediction and ability to account implicitly for selection [17]. In addition, ssGBLUP presented dispersion parameters closest to the optimal value of one, and it demonstrated the consistency of this estimator of this type of model. Indeed, dispersion represents regression of EBV from whole to partial data, thus making the model less affected by the addition/subtraction of information, and therefore the best model to be applied.

Our finding supports the use of genomic data, and in particular the use of ssGBLUP, as the new model for the routinary genetic evaluation on selected bulls of a local breed, the Rendena cattle. In local breeds, genomic information has mainly been used to assess genetic variability or to study specific biological pathways underscoring peculiar traits such as [56]. As mentioned above, this is a first study investigating the impact of genomic information on selection in indigenous breeds [57]. We focused on performance test traits because of the antagonistic relation to milk traits, but genomic selection can be successfully applied also for other traits, depending on the amount of phenotypic and genomic information available. Notably, the increase in the accuracy of selection can impact the economic value of the breed [58], which is a pragmatic and effective strategy to guarantee the conservation of local breeds.

The present study shows that genomic imputation and the combination of genotyped and non-genotyped data through ssGBLUP could be a cost-efficiency strategy, compensating for the limited genotype information available on local breeds. This could make genomic selection for limited populations an appealing strategy, as it already is in more cosmopolite breeds such as Holstein [59] However, the LR cross-validation demonstrated that accuracy increased only in young bulls with genotypes, while the accuracy of non-genotyped animals was only marginally higher than that obtained with the PBLUP in the subgroups of individuals with a genotyped close. For this reason, we would recommend, to keep increasing selection accuracy, that a majority of animals for each performance test cycle should still continue to be genotyped.

## 5. Conclusions

All models that included genomic data presented higher accuracy and reliability than the ones using only kinship information. These two estimators were particularly higher in models in which high heterogeneous variances among SNPs had been assumed; however, the same models presented under-dispersion and higher bias, and for that reason, they can be discarded as models to be used in the selection. Models with “best properties” can be identified in the ssGBLUP or in the WssGBLUP, in which weighting strategies presented less shrinkage. Although these two models presented similar proprieties, ssGBLUP could be chosen as the “best” model, because it was neither under- nor over-dispersed, presenting appropriate properties for long-term selection. In conclusion, the present study demonstrated how the use of genomic data in addition to ssGBLUP can lead to a better prediction of genetic effects even with a modest amount of molecular data, as typically happens in local populations. Therefore, we demonstrated how genomic data can be a suitable tool for breeding selection scenarios in local cattle breeds such as Rendena, guaranteeing the competitiveness and thus the conservation of the breed through its improvement of selection’s accuracy.

## Figures and Tables

**Figure 1 animals-11-01815-f001:**
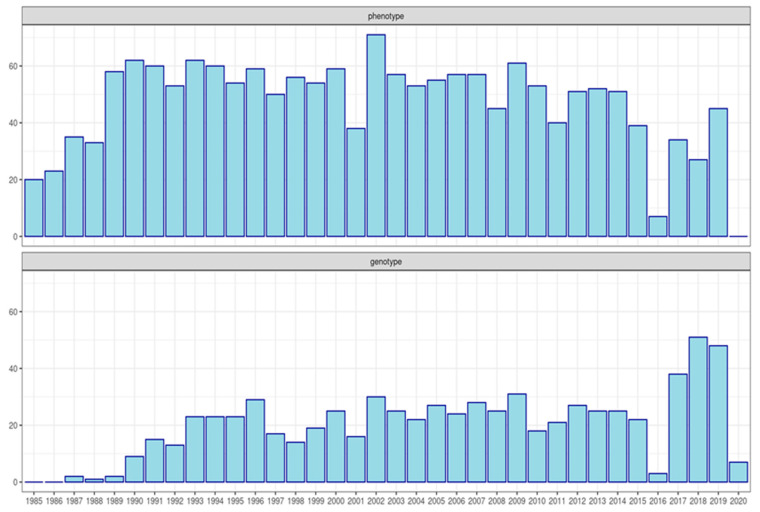
Number of animals with the phenotype (**above**) and number of animals with the genotype (**bottom**) for all animals used in genetic or genomic prediction. X-axes represent the birth years and y-axis the number of animals per year.

**Figure 2 animals-11-01815-f002:**
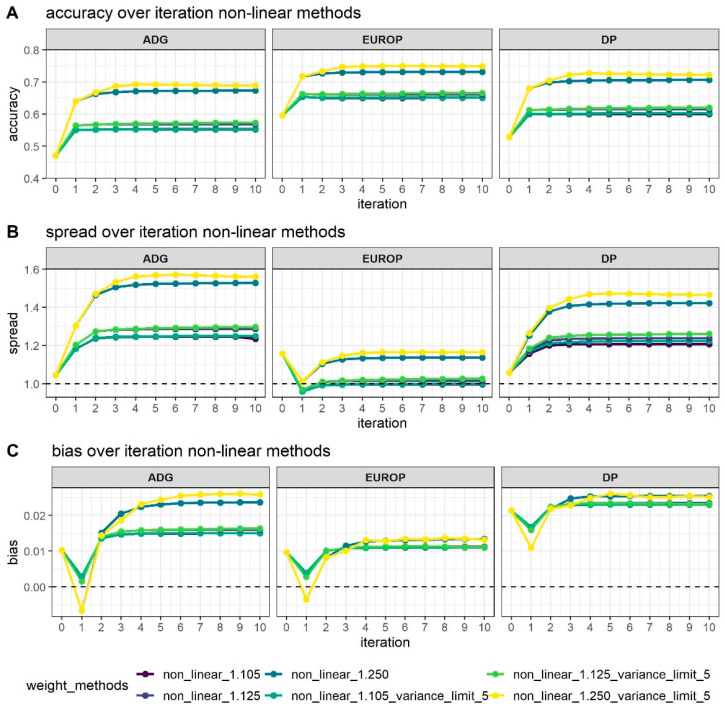
Accuracy (**A**), dispersion (**B**), and bias corrected by genetic standard deviations (**C**) of breeding value estimated using different weighting strategies along the 10 iterations process of the algorithm used in WssGBLUP. The dotted line in Graphs B and C represents the expected value.

**Figure 3 animals-11-01815-f003:**
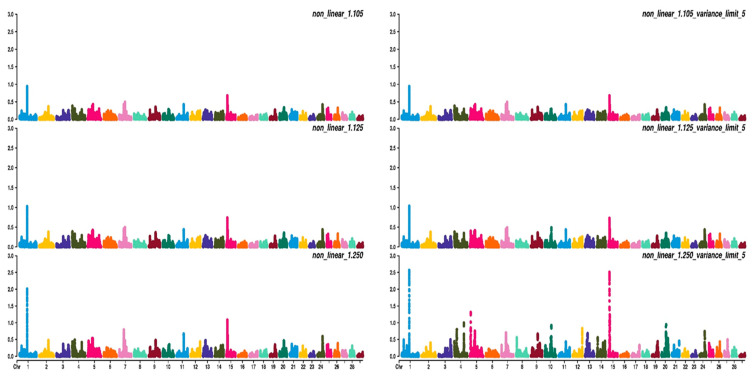
Manhattan plots for average daily gain (ADG) using different WssGBLUP strategies in iterations equal to 10; y-axes represent the percentage explained by each SNP. Variance explained was calculated with a sliding window approach.

**Figure 4 animals-11-01815-f004:**
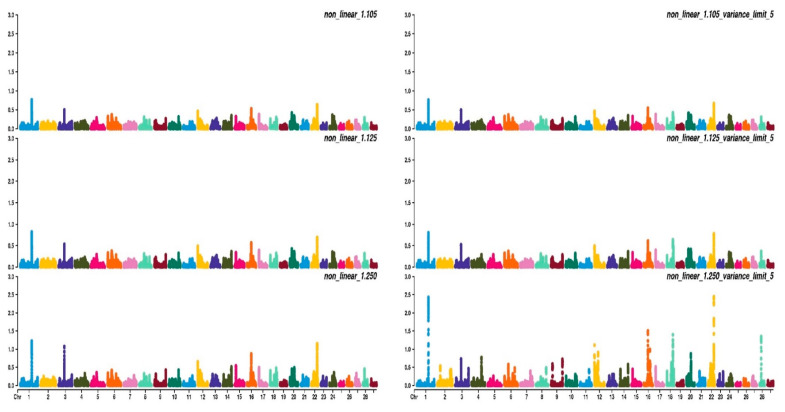
Manhattan plots for fleshiness score (EUROP) using different WssGBLUP strategies in iteration equal to 10; y-axes represent the percentage explained by each SNP. Variance explained was calculated with the sliding window approach.

**Figure 5 animals-11-01815-f005:**
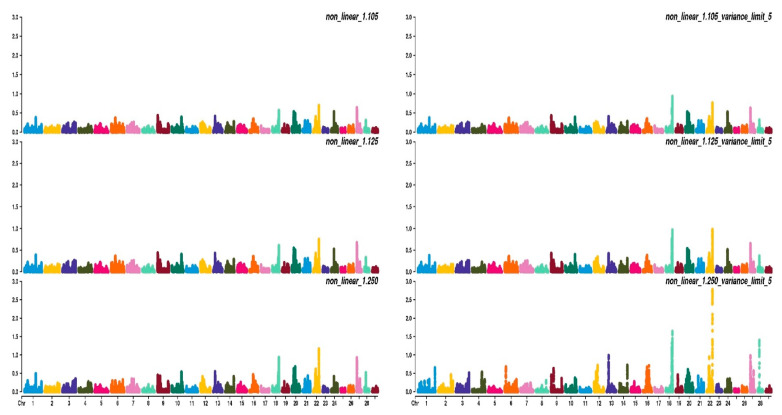
Manhattan plots for dressing percentage (DP) using different WssGBLUP strategies in iterations equal to 10; y-axes represent the percentage explained by each SNP. Variance explained was calculated with the sliding window approach.

**Figure 6 animals-11-01815-f006:**
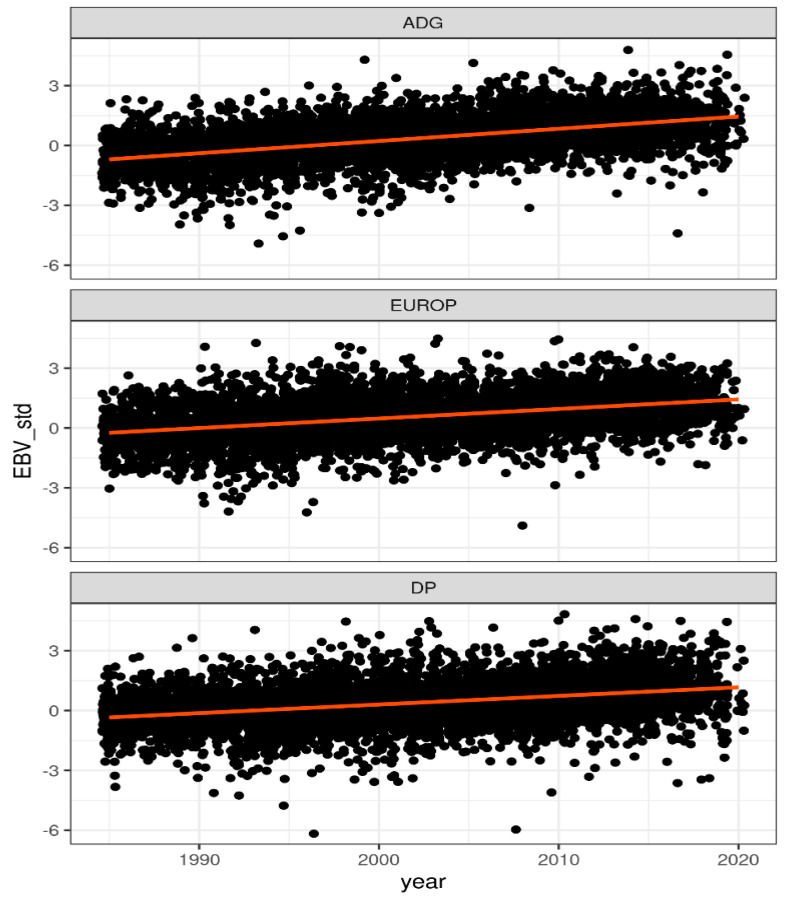
Standardize genetic progress per year: x-axis indicates birth year of animals and y-axis the standardized EBV, from 1985 (when performance test started) to 2020 (current data).

**Table 1 animals-11-01815-t001:** Descriptive statistics of the target phenotypic data obtained from Rendena young bulls under performance test.

Taits ^1^	Number ^2^	Mean	CV %	Min	Max
ADG (kg/d)	1691 (690)	1024.00	12.06	474.00	1562
EUROP (points)	1691 (690)	99.05	3.84	80.00	111.10
DP (points)	1691 (690)	54.18	1.74	50.00	57.70

^1^ ADG = Average daily gain; EUROP = in vivo fleshiness score, DP = in vivo estimate of dressing percentage. ^2^ Number of animals with records and (genotype).

**Table 2 animals-11-01815-t002:** Mean of genetic (upper diagonal) and residual (lower diagonal) correlations and heritability (diagonal) between traits in the Rendena population, estimated with PBLUP. Numbers in parenthesis are the lower and the upper 95% highest posterior density.

	ADG	EUROP	DP
ADG	0.335 (0.204 ± 0.335)	0.364 (0.100 ± 0.597)	0.398 (0.148 ± 0.6315)
EUROP	0.572 (0.660 ± 0.742)	0.304 (0.174 ± 0.446)	0.981 (0.962 ± 0.997)
DP	0.613 (0.517 ± 0.702)	0.792 (0.753 ± 0.836)	0.392 (0.248 ± 0.541)

ADG = Average daily gain, EUROP = and in vivo fleshiness score CY, DP = in vivo estimate of dressing percentage.

**Table 3 animals-11-01815-t003:** Mean of genetic (upper diagonal) and residual (lower diagonal) correlation and heritability (diagonal) between traits in Rendena population, estimated with ssGBLUP. Numbers in parenthesis are the lower and the upper 95% highest posterior density.

	ADG	EUROP	DP
ADG	0.313 (0.223 ± 0.489)	0.385 (0.153 ± 0.597)	0.392 (0.160 ± 0.622)
EUROP	0.651 (0.651 ± 0.718)	0.345 (0.216 ± 0.487)	0.985 (0.961 ± 0.999)
CY	0.616 (0.530 ± 0.671)	0.790 (0.753 ± 0.826)	0.396 (0.250 ± 0.530)

ADG = Average daily gain, EUROP = and in vivo fleshiness score CY, DP = in vivo estimate of dressing percentage.

**Table 4 animals-11-01815-t004:** Accuracy, bias, dispersion (Disp.), and reliability (Rel.) and adjusted increased of accuracy (Incr_adj) of estimated breeding values under different models: pedigree BLUP (PBLUP), single-step genomic BLUP (ssGBLUP), and weight single-step with bias value closet to optimal value (WssGBLUP_1) and weight single-step with highest accuracy; for average daily gain (ADG), EUROP, and dressing percentage (DP).

Trait	Model	Accuracy	Bias	Disp.	Rel.	Incr_adj
ADG	PBLUP	0.366	−0.040	1.140	0.060	-
	ssGBLUP	0.472	0.010	1.045	0.117	45.10%
	WssGBLUP_1	0.551	0.003	1.182	0.127	45.10%
	WssGBLUP_2	0.693	0.020	1.562	0.206	49.21%
EUROP	PBLUP	0.509	−0.009	0.902	0.081	-
	ssGBLUP	0.596	0.009	1.100	0.124	39.98%
	WssGBLUP_1	0.653	0.004	0.958	0.135	39.98%
	WssGBLUP_2	0.749	0.014	1.165	0.192	45.17%
DP	PBLUP	0.464	−0.021	1.114	0.114	-
	ssGBLUP	0.528	0.021	1.056	0.158	26.70%
	WssGBLUP_1	0.600	0.017	1.156	0.184	27.40%
	WssGBLUP_2	0.727	0.025	1.468	0.277	33.90%

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
