# Peer review of "Genomic Prediction in Local Breeds: The Rendena Cattle as a Case Study"

_animals, 2021, doi:10.3390/ani11061815_

Round 1
Reviewer 1 Report
Dear authors,
thank you for your research. The investigation of methods to integrate genomic data for the evaluation of local breeds is very important. This the first manuscript that I have reviewed that it is really well written and explained. I have noted a few minor things:
1- page 12 line 8: where you have table 3 I think you have meant table 4
2- page 14, line 35 the same issue
3- page 15 line 106 where it is stated reason I suggest to change to reasons
I have selected "English language and style are fine/minor spell check required" only because" the form required the selection of one of the options. But I think the English is excellent.
Author Response
Review 1.)
Dear authors,
thank you for your research. The investigation of methods to integrate genomic data for the evaluation of local breeds is very important. This the first manuscript that I have reviewed that it is really well written and explained. I have noted a few minor things:
AU: We sincerely thank the reviewer for their work and for their kind words.
- page 12 line 8: where you have table 3 I think you have meant table 4
AU: We apologize for oversight, corrections have been made, thank you.
2- page 14, line 35 the same issue
AU: We apologize for the same oversight, corrections have been made, thank you.
3- page 15 line 106 where it is stated reason I suggest to change to reasons
AU: Thanks for notice it, changes have been made please, see line 106 pag. 15
Reviewer 2 Report
The manuscript: “Genomic prediction in local breeds: the Rendena cattle as a case study” is an excellent manuscript. The analyses are performed well; the sample size is reasonable. The manuscript is well written, and the results are well discussed.
I wonder why the authors did not include GBLUP in the comparison. Some information is not clear in Table 3,
Minors:
Line 17: Better than what?
Line 29: The authors might specify what is LR?
Line 56-57: Which phenotypes have been estimated using PBLUP? Please extend it.
Table 2 -3: Please specify the values in (), and why did not the authors give the mean and SE for the correlation?
Figure 2 title: Change” methos” to methods.
The legend for the line is too small; please increase it.
Line 281, 294-295: Write full name as heritability instead of h2; it is not necessary. Otherwise using h2
Page 15: in [49] might write as in the Cesarani et al. [49]
Line 135: single-step GBLUP, change to ssGBLUP (page 16)
Author Response
The manuscript: “Genomic prediction in local breeds: the Rendena cattle as a case study” is an excellent manuscript. The analyses are performed well; the sample size is reasonable. The manuscript is well written, and the results are well discussed.
I wonder why the authors did not include GBLUP in the comparison. Some information is not clear in Table 3,
AU: We thank the reviewer for their appreciative words and their thorough evaluations. Regarding the exclusion of GBLUP as a model for genomic prediction, we have decided to omit it, because we aim to mimic a true herd selection scheme, and since only a fraction of animals have been genotyped, it is important to ensure an index of selection for this animal, and the ssGBLUP makes it possible to estimate the EBV of these animals in a simpler and more precise way. Additionally, many studies have reported ssGBLUP to be more accurate than BLUP, so it seemed redundant to use GBLUP. However, we thank you for this careful clarification.
Minors:
Line 17: Better than what?
AU: We thank the reviewer to point out this carelessness, please see line 17
Line 29: The authors might specify what is LR?
AU: LR was the cross validation methods used in that study, LR has been replaced with "cross validation" which is in accord with the general sense of the abstract
Line 56-57: Which phenotypes have been estimated using PBLUP? Please extend it.
AU: We apologize for implying this, phenotypes estimated with BLUP was added please see line 56-57
Table 2 -3: Please specify the values in (), and why did not the authors give the mean and SE for the correlation?
AU: Since the common approach used by the Bayesian method is to report higher posterior density interval (5-95) we have added in the description of the table "The numbers in brackets are the lower and upper posterior density of the highest 95%". Thanks for noticing it.
Figure 2 title: Change” methos” to methods.
The legend for the line is too small; please increase it.
AU: Thanks for the observation, figure 2 has been improved.
Line 281, 294-295: Write full name as heritability instead of h2; it is not necessary. Otherwise using h2
AU: Thanks for notice that, h2 was changed with h2
Page 15: in [49] might write as in the Cesarani et al. [49]
AU: Thanks for the observations, citation was changed.
Line 135: single-step GBLUP, change to ssGBLUP (page 16)
AU: Thanks for the observations, we substituted single-step GBLUP with ssGBLUP